# Generalization-Based Acquisition of Training Data for Motor Primitive Learning by Neural Networks

**Zvezdan Lončarević** [1,2,*], **Rok Pahič** [1,2], **Aleš Ude** [1,3] and **Andrej Gams** [1,2]

1 Humanoid and Cognitive Robotics Lab, Department of Automatics, Biocybernetics and Robotics, Jožef Stefan Institute, Jamova 39, 1000 Ljubljana, Slovenia; rok.pahic@ijs.si (R.P.); ales.ude@ijs.si (A.U.); andrej.gams@ijs.si (A.G.)
2 Jožef Stefan International Postgraduate School, Jamova 39, 1000 Ljubljana, Slovenia
3 Faculty of Electrical Engineering, University of Ljubljana, Tržaška 25, 1000 Ljubljana, Slovenia
* Correspondence: zvezdan.loncarevic@ijs.si

**Abstract:** Autonomous robot learning in unstructured environments often faces the problem that the dimensionality of the search space is too large for practical applications. Dimensionality reduction techniques have been developed to address this problem and describe motor skills in low-dimensional latent spaces. Most of these techniques require the availability of a sufficiently large database of example task executions to compute the latent space. However, the generation of many example task executions on a real robot is tedious, and prone to errors and equipment failures. The main result of this paper is a new approach for efficient database gathering by performing a small number of task executions with a real robot and applying statistical generalization, e.g., Gaussian process regression, to generate more data. We have shown in our experiments that the data generated this way can be used for dimensionality reduction with autoencoder neural networks. The resulting latent spaces can be exploited to implement robot learning more efficiently. The proposed approach has been evaluated on the problem of robotic throwing at a target. Simulation and real-world results with a humanoid robot TALOS are provided. They confirm the effectiveness of generalization-based database acquisition and the efficiency of learning in a low-dimensional latent space.

**Keywords:** autoencoders; robot learning; statistical generalization; dimensionality reduction

## 1. Introduction

Robot learning, a process where the robot improves its performance by executing the desired task many times to update the principal skill representation, is one of the main technological enablers that can take robots into unstructured environments [1,2]. Nevertheless, robot learning can be a complicated and lengthy process, which requires numerous iterations, trials, and repetitions, all of which might not be safe for the robot or its immediate environment. This is specifically the case when the robot needs to learn a new task from scratch—the search space is simply too large [3]. This is also the case for monolithic problems where only one type of solution is possible (in our practical example, only one way of throwing) [4]. Intuitively, by reducing the dimension of the search space for learning, more successful autonomous learning algorithms can be implemented [5].

The learning process can be made more efficient in different ways. For example, learning by demonstration (LbD) may provide an initial approximation, which is used to initiate the selected learning algorithm [6]. As a human expert can demonstrate the task only for the finite number of states of the real world, the accumulated robot knowledge cannot be applicable in all possible states unless the robot can generalize from it [7]. Statistical generalization can be and has been applied for generation of actions from a set of example executions of a task [8–10]. This generation is possible only if the example executions are related through a known set of parameters, e.g., a parametrized goal of the action, and if they are continuous. For example, in an action of reaching for an object, one cannot

easily generalize between reaching for an object from the left and from the right, but only separately between examples of reaching from the same class (same side) of reaching movements [11].

If the action computed by statistical generalization does not provide a policy that successfully executes the task in the current state of the environment, it needs to be improved. This can be done with Reinforcement Learning (RL), which provides a methodology and toolkits for the design of hard-to-engineer, complex behaviors [12]. Policy search is a sub-field of Reinforcement Learning [13] with the focus on finding parameters for a given policy parametrization. In policy search, no analytical model is available [14], but learning a policy is often easier than learning an accurate forward model [13]. The drawback of this approach is that each sampled instance of execution needs to interact with the robot, which can be time-consuming and challenging in practice—every example movement takes some time to execute. Even though parametrized policies scale continuous actions in high-dimensional spaces for RL, the dimensionality of the search space is often still beyond the range of practical applications. This is often referred to as the curse of dimensionality [15].

The search space thus needs to be reduced as much as possible. One possibility is to project data onto the vector space spanned by basis vectors defined by the variance of the data, known as Principal Component Analysis (PCA) [16]. Another possibility is to use latent space of autoencoders. Autoencoders (AEs) are artificial neural networks used to learn efficient data encoding in an unsupervised manner. They push data through the layers of the neural network, and the layer with the smallest number of neurons—the latent space—can be of smaller dimensionality than the input data. While PCA provides a linear transformation of the data, AEs provide a nonlinear transformation, which has been shown to be more appropriate for the nonlinear world we live in [17]. In order for their respective transformations to be trained, both PCA and AEs require a database of actions. There is always some loss of information when parameterizing actions. It is therefore important to compute high-quality latent spaces that provide good representation of robot movements and contain the optimal movement for the given task.

### 1.1. Problem Statement

It was shown in the literature that learning in autoencoder latent space outperforms learning in the full policy space [5] and learning in PCA latent space [17,18]. This paper explores the usage of statistical generalization to create the database, i.e., training samples required to train autoencoder neural networks for dimensionality reduction, thus reducing the required real-world policy executions. Additionally, the paper explores the required number of training samples and dimension of autoencoder latent space.

To corroborate the results, we evaluated the proposed training sample acquisition method and investigated the required number of training samples for real-world learning of ball throwing at a target with a TALOS humanoid robot [19]. The experimental setup is depicted in Figure 1.

### 1.2. Related Work

As outlined in the introduction, the topic of this paper is learning in a reduced space, addressing the issue of how to obtain a suitable action database. It has been shown that learning of robot actions can be more effective in a smaller search space [15]. Therefore, different dimensionality reduction techniques were used to reduce the search space, for example, PCA, discriminant function analysis—such as linear discriminant analysis (LDA), kernel PCA, and AE. The first two are linear in nature, while kernel PCA and AE are not. These methods were applied in various robotic applications. As an example, PCA was used to reduce the amount of training data for learning the behavior of an articulated body in [20] and to obtain optimal robot motion in real-time [21]. These two works both say that obtaining a suitable database and working with high amounts of data are among the key problems. LDA, on the other hand, is typically applied for classification, but not necessarily better than PCA [22].

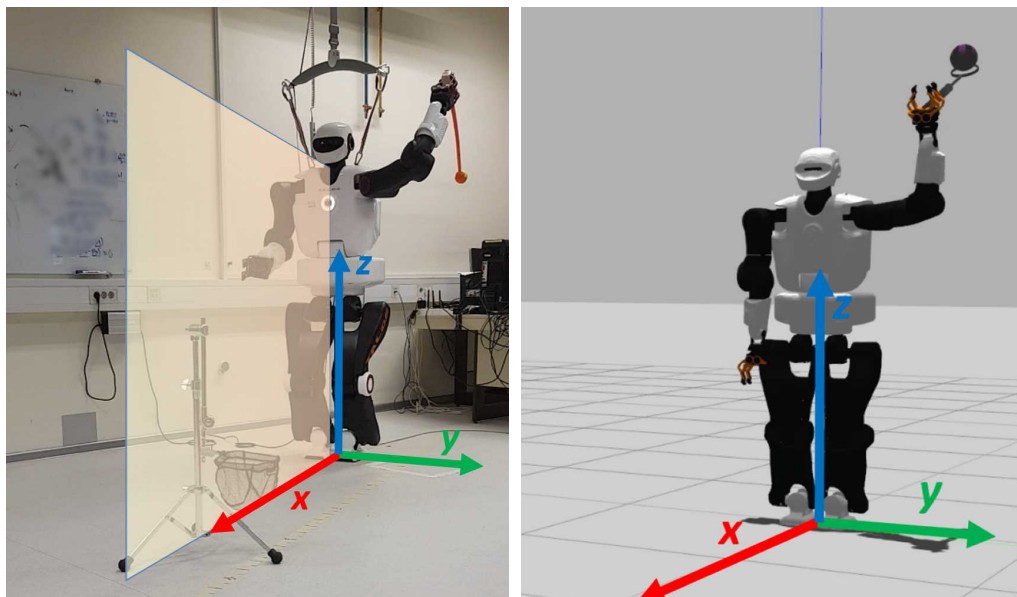

**Figure 1.** TALOS humanoid robot in the throwing posture with the projected saggital ($x - z$) plane (**left**); the robot in GAZEBO simulation (**right**).

Linear approximation of real-world actions might not be the best for the nonlinear world. It has been shown that AEs work much better than PCA to reduce the dimensionality of the data [17]. Similar results were reported in [18], where generalization of throwing actions in either PCA latent space or AE latent space were compared, with the database acquired in simulation. Kernel PCA [23] can provide for a nonlinear reduction of dimensionality of data. Among the issues that need to be addressed to implement kernel PCA are the storage of a large kernel matrix and the selection of the nonlinear kernel. The utilization of different kernels was investigated for modeling of humanoid robot motions in [24]. Given that we are dealing with a nonlinear task in a nonlinear world, we chose autoencoders for dimensionality reduction.

All dimensionality reduction techniques require a database. It is often not feasible to record thousands of robot actions that are needed to train neural networks. To illustrate, in order to learn proper grasping techniques, close to a million of repetitions were carried out with a cluster of robots [25]. The question of database acquisition was tackled also in other settings, e.g., for statistical generalization that typically requires less data than neural network training. In [26], the authors investigated how many example executions and in which order they need to be added to enable meaningful generalization. Pahič et al. [18] studied how many database entries are required for generalization of real-world actions using a simulated database. Other instances of database learning for generalization include the work in [27], where the authors autonomously added database entries after generalization and adaptation. Similar ideas were investigated in [28,29]. In this paper, we use a different approach. We first record a small number or real-world actions and generalize to a larger set. We use this larger set to train autoencoder, and then implement learning methodologies in its latent space. Action representations and Reinforcement Learning (RL) are discussed in, for example, [5,18,30,31]. Furthermore, action generalization in feature space of a quadruped robot is discussed in [32].

RL and especially Deep Reinforcement Learning (DRL) achieved great results in simulated environments [33,34]. Nevertheless, all these achievements depend on the ability to collect large amounts of accurate data. For the robots that are equipped only with noisy sensors data and are supposed to work in everyday and ever-changing environments, the application of such learning algorithms might be difficult. A promising method is to train control policies in simulated environments, where data generation is safe and convenient, and then transfer the learned policies to the real world [35]. This process is called transfer learning (TL). Several methods were developed in order to bridge the gap

between the simulation and real-world [36,37], but it still remains an open topic of research. Creating the simulated environment for some more complex tasks is often not feasible. This is the reason why in this paper we examine the possibility of using generalization over a small set of real-world example executions to acquire larger datasets instead of simulation.

## 2. Methods and Data Acquisition for Learning

In the following, we first describe the proposed methodology to solve the above-stated problem. We start by defining the learning space and then explain how to implement dimensionality reduction of the learning space using autoencoders. We further discuss database construction by the application of statistical generalization methodologies.

These techniques are needed to derive the main contributions of this paper: a new methodology for obtaining samples trajectories grounded in the real-world data by means of statistical generalization and the analysis of the required number of training samples for autoencoder-based dimensionality reduction. In short, this paper proposes the following methodology to improve the accuracy of robot behavior and the speed of RL convergence. We first record a small number of real-world actions and generalize to a larger set. We then use this larger set to train an autoencoder. The latent space of the autoencoder is smaller than the full action representation space, which provides for a faster convergence of Reinforcement Learning. Because the data for autoencoder training are grounded in a small set of real-world actions, it is also closer to real-world actions than an autoencoder trained with only simulated data. As explained above, the consequence is that we reduce the amount of real-world robot executions to acquire a dataset suitable for autoencoder training and then learn faster because RL is performed in a smaller search space defined by the latent space of the autoencoder.

### 2.1. Dimensionality Reduction of Policy Parameter Space with Autoencoders

Let us consider parametric policies $\Pi(\boldsymbol{\theta})$, where $\boldsymbol{\theta} \in \boldsymbol{\Theta}$ are the policy parameters and $\boldsymbol{\Theta} \subset \mathbb{R}^d$ is a $d$ dimensional space formed by all valid policy parameters $\boldsymbol{\theta}$. Kober and Peters [15] noted that the dimensionality of $\boldsymbol{\Theta}$ can be problematic when learning optimal policies for a given task. Intuitively, learning should become easier and faster to perform if we could implement it in a lower-dimensional parametric space. For this purpose, we must first embed all valid policies for the given task into a lower-dimensional parametric space. This is often possible because the space of all valid policies for the given task forms a low-dimensional manifold in the full parameter space $\boldsymbol{\Theta}$ [9].

As shown in [30], deep autoencoder neural networks enable dimensionality reduction while retaining the most pertinent information in the low-dimensional policy representation. The autoencoder neural network is trained so that its input data matches its output data as precisely as possible. The input data are pushed through the network layers until they reach the bottleneck—the layer with the least number of neurons, also called the latent space. We denote the values of the neurons in the bottleneck layer by $\boldsymbol{\theta}^{\mathrm{AE}}$. The part from the input to the bottleneck is called the encoder part. The second part of the autoencoder, called the decoder, pushes data from the bottleneck through expanding layers, so that the outputs $\widetilde{\boldsymbol{\theta}}$ of the autoencoder match the input data $\boldsymbol{\theta}$ as closely as possible. Thus, the encoder and the decoder functions are nearly inverse $\mathbf{F}_{\mathrm{dec}} \approx \mathbf{F}_{\mathrm{enc}}^{-1}$. The dimensionality of the data is now defined by the number of neurons in the bottleneck layer—the latent space (also referred to as code). Its dimensionality $d_{\mathrm{AE}}$ is usually significantly lower than the dimensionality of the original space, i.e., $d_{\mathrm{AE}} < d$.

Thus, given an input data, the values of the neurons in the latent space are affected by the activation functions of the other hidden layers. These activation function can be, for example, the hyperbolic tangent (tanh), the sigmoid function, or a rectified linear unit [38]. For the tanh function, the output values of neurons at layer $n$ are given by $\boldsymbol{h}_n = \tanh(\mathbf{W}_n \boldsymbol{h}_{n-1} + \mathbf{b}_n)$, where $\boldsymbol{h}_{n-1}$ are the input values for the neurons at layer $n$ and $\mathbf{W}_n$, and $\mathbf{b}_n$ are autoencoder parameters. Their values are determined by training.

To train the AE network, that is to compute the appropriate autoencoder parameters $\zeta^\star$, we minimize

$$\zeta^\star = \arg\min_{\zeta} \frac{1}{N_s} \sum_{i=1}^{N_s} L(\zeta; \boldsymbol{\theta}_i, \mathbf{F}_{\text{dec}}(\mathbf{F}_{\text{enc}}(\boldsymbol{\theta}_i))), \quad (1)$$

where $\zeta = \bigcup_n \zeta_n$, $\zeta_n = \{\mathbf{W}_n, \mathbf{b}_n\}$, combines all the parameters defining the autoencoder network, $\boldsymbol{\theta}_i$ are the example policy parameters, $L$ is the Euclidean distance between the input and output of the autoencoder network, and $N_s$ is the number of samples, i.e., in our case this is the number of trajectories. We use stochastic gradient descent to train the network, thus the results of optimization depend on the initialization of the optimization process. For this reason, the network is trained several times using different initial parameters. The results are then averaged. The training of each autoencoder network is terminated if the results do not improve in 60 consecutive validation steps. This number was determined empirically.

Upon training of the network, the latent space representation of a robot movement, which is specified by policy $\Pi(\boldsymbol{\theta})$, can be computed by applying the encoder part of the network,

$$\boldsymbol{\theta}^{\text{AE}} = \mathbf{F}_{\text{enc}}(\boldsymbol{\theta}). \quad (2)$$

In the same manner, the mapping from the autoencoder latent space $\boldsymbol{\theta}^{\text{AE}}$ back to policy parameters is given by the decoder part of the network,

$$\widetilde{\boldsymbol{\theta}} = \mathbf{F}_{\text{dec}}\left(\boldsymbol{\theta}^{\text{AE}}\right). \quad (3)$$

### 2.2. Database Acquisition by Generalization

As in all neural networks, autoencoders also require a large amount of samples for training. Obtaining such an amount of samples with a robot is a time-consuming job that causes too much wear and tear of the equipment and might even damage the robot itself. One way to address these issues is to gather data in simulation. However, there is always a discrepancy between the simulation dynamics and the real system. The main contribution of this paper is a way to synthetically form the training samples without the need for many thousands of repetitions with a real system.

To compute the autoencoder network defining the AE-based latent space, we must first acquire a training dataset of robot motion trajectories

$$\{\boldsymbol{\theta}_i\}_{i=1}^{N_s}, \quad (4)$$

where $N_s$ is the number of trajectories and $\boldsymbol{\theta}_i$ are the parameters representing the $i$-th trajectory. The trajectories can be acquired either in simulation or in the real world. The former does not require task executions with a real robot but might result in an inaccurate description of the task because no model is fully accurate. Moreover, it is sometimes difficult to construct mathematical models needed to generate simulated data. The latter will faithfully describe the behavior, but requires many task executions with a real robot, which is very time-consuming, to the point that it might not be practically feasible.

In this paper, we propose a new approach to generate data for learning autoencoders using statistical generalization methods, e.g., Gaussian Process Regression (GPR) [39]. The idea is to start by collecting a relatively small number of task executions with a real robot and then apply GPR to generate a larger set of synthetic data based on the available real data. Among many possible statistical learning methods, we selected GPR because it has been demonstrated [39] that in many practical problems GPR achieves better performance than other methods for robot learning. For example, GPR has been shown to be effective when estimating the inverse dynamics of a seven DOF robot arm [40].

Let us assume that the robot performed a relatively small number of tasks. We denote the associated movements and task descriptors as $\boldsymbol{\theta}_i$ and $\boldsymbol{q}_i$, respectively. From these data we form a dataset

$$\mathcal{D} = \{\boldsymbol{\theta}_i, \boldsymbol{q}_i\}_{i=1}^{N_s}. \tag{5}$$

New data points can be generated by selecting a new desired task descriptor $\boldsymbol{q}_d$ and computing the associated motion policy by GPR, i.e.,

$$G(\mathcal{D}) : \boldsymbol{q}_d \to \boldsymbol{\theta}_d. \tag{6}$$

For the sake of completeness, we provide the formulas to implement GPR in Appendix B.

Note that it is not necessary to execute the task with the robot or to simulate the task execution in a simulation system when generating new data couples $\boldsymbol{\theta}_d$, $\boldsymbol{q}_d$. Thus new data can be computed very efficiently. The robot only executes a few shots at different targets while the rest of the data are generated by GPR. Our assumption, which is confirmed in our experiments, is that the training data obtained with this approach will perform as well or better than the data obtained in simulation. Consequently, a better initial approximation reduces the number of RL iterations.

## 3. Experimental Setup and Protocol

In our evaluation experiments, we first studied the required size of the database of example task executions, which was used to compute latent spaces. The goal was to represent motion policies in latent spaces without loosing much information. We tested several networks with different dimensions of the latent space and trained them on different number of actions. The following sections provide a detailed description of experiments and confirm that our method for the generation of data for latent space computation results in good latent space representations that significantly improve the overall performance of robot learning.

### 3.1. Task and Robot

We performed the experimental evaluation of the proposed methodology by implementing the task of throwing a ball at a target (basket) with a robot arm. We assume that the orientation of the robot in the horizontal plane ($x - y$, see Figure 1) is correct. This assumption does not reduce the generality because correcting the orientation in the horizontal plane can take place before the actual throwing. Thus, the throwing is a planar problem in the vertical (saggital) plane of the robot ($x - z$, see Figure 1). The target of throwing, which in our case is a basket, is located in the saggital plane and can be described by two parameters, i.e., distance and height from the robot's base.

We used the full-sized humanoid robot TALOS in our experiments. They were performed both in Gazebo simulation [41] and on a real robot. Out of 32 robot DOFs, we used 7 DOFs of its left arm to generate the throwing motions. The movements where specified in task space and generated using inverse kinematics. All 6 DOFs in the task space were controlled. Inverse kinematics was implemented using a null-space controller to keep the joints as much as possible in the middle of the joint limits. To extend the range of throwing, the robot held in its hand a passive elastic throwing spoon. The humanoid robot was standing upright in a static posture, which ensured that the center of mass projection was between the two feet. The throw itself was considered as a perturbation and no active measures were undertaken to counter its effects on the stability.

The robot held the aforementioned spoon with its hand, but the ball was not firmly attached to the spoon. Thus, during the throwing action, the ball detached itself on its own from the holder once the hand motion started to slow down, i.e., when the arm acceleration became negative. This approach was taken both in dynamic simulation and on the real robot. Note that this is different from humans who usually firmly hold the ball during the throwing movement and determine the release point by letting the ball go. An image sequence of a successful throw in simulation is shown in Figure 2.

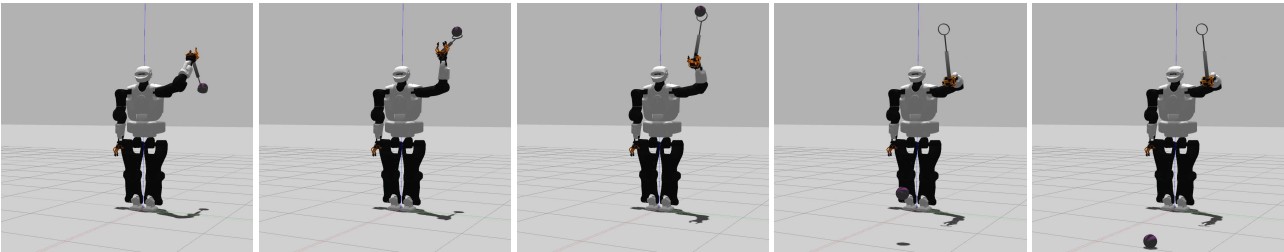

**Figure 2.** Image sequence of a throw in GAZEBO simulation environment.

### 3.2. Policy Parameter Space and Latent Space

In our experiments, we represented a throwing trajectory with a Cartesian space dynamic movement primitive (CDMP) [42]. Thus, the full parameter space of throwing trajectories is defined by the free CDMP parameters. CDMPs encode the motion of each robot degree of freedom with a nonlinear dynamic system. While positions are encoded in the same manner as, for example, joint space trajectories (see [43]), Cartesian space orientations in the form of unit quaternions are encoded differently in order to maintain the unit norm [42]. The free CDMP parameters consist of the starting position $p^0$ and orientation $o^0$, weights defining the dynamic of motion for the position and orientation $w_k^p$, $w_k^o \in \mathbb{R}^3$, $k = 1, \ldots, N$, trajectory duration $\tau$, and the final (desired) position $g^p$ and orientation $g^o$ of the robot arm motion. If, for example, the number of weights is $N = 25$ (this is usually sufficient for good accuracy, see in [9]), we obtain a 165 dimensional parameter space. For clarity, we provide a short recap of CDMPs in Appendix A. Further details are available in [42]. The application of neural networks for the generation of dynamic movement primitives has been analyzed in detail in our previous work [44].

In our experiments, throwing is a planar problem in the $x - z$ plane. Thus, only three Cartesian space degrees of arm motion must be specified: the positional motion along the $x$ and $z$ axis and the rotational motion around $y$ axis. The other three degrees of freedom are fixed. In our experiments, we also kept constant the initial and goal position of the arm motion (position and orientation of the throwing spoon). Thus, the throwing movement always starts and ends in the same poses. The relevant DOFs and their parameters are the following.

$$\boldsymbol{\theta}^{\text{CDMP}} = \left[ \boldsymbol{w}_x^T, \boldsymbol{w}_z^T, \boldsymbol{w}_{rot_y}^T, \tau \right]^T. \tag{7}$$

By setting the number of weights to $N = 25$, the dimension of the policy space becomes $d_{\text{CDMP}} = 76$.

We applied an autoencoder neural network to further reduce the dimensionality of the full CDMP parameter space. The parameters specified in Equation (7) represent the inputs and outputs of the autoencoder network. The autoencoder structure depends on the task and the manner in which the data are encoded. It needs to be chosen in such a way that the difference between training inputs and outputs is as low as possible. Typically, the autoencoder network structure is empirically determined by the network designer, keeping in mind the potential of vanishing or exploding gradients [45].

In our experiments, the autoencoder network was a fully connected, 5-layered neural network with 76-20-L-20-76 neurons. This means that there were 3 hidden layers with 20, L, 20, neurons. In order to determine the appropriate latent space layer, we varied L between 2 and 10. Section 4.1 provides the results of the autoencoder network approximation depending on the dimension of the latent space and the number of samples used in the training process. Figure 3 shows an illustration of the used autoencoder structure, but keep in mind that we varied the number of latent space neurons.

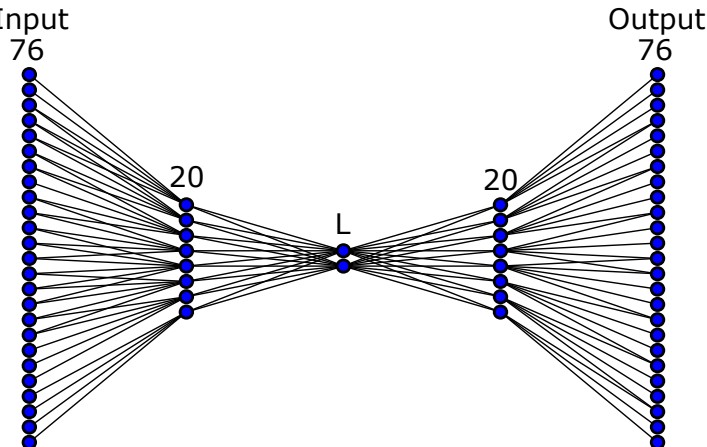

**Figure 3.** Illustration of the selected AE neural network structure. Dimension of the latent space was varied between 2 and 10.

### 3.3. Database for AE Training

For evaluation purposes we collected several databases. The first database was used to assess the accuracy of AE training. It should be noted that just to assess the accuracy of AE training, arbitrary throwing trajectories can be used as input data, as only the control policy parameters are important for this purpose. We computed 800 simulated throwing trajectories and formed the database

$$\mathcal{D}^{\text{AE}} = \left\{ \boldsymbol{\theta}_j^{\text{CDMP}}, \boldsymbol{q}_j^{\text{AE}} \right\}_{j=1}^{800}. \tag{8}$$

Different random subsets of $\mathcal{D}^{\text{AE}}$ were used to evaluate the accuracy of AE training.

For the next database, we computed 625 evenly distributed targets $\{\boldsymbol{q}_j^g\}_{j=1}^{625}$ (located in-between targets $\boldsymbol{q}_j^{AE}$) and applied generalization to compute the appropriate throwing policies

$$\mathbf{G}(\mathcal{D}^{\text{AE}}) : \boldsymbol{q}_j^g \mapsto \boldsymbol{\theta}_j^{\text{CDMP},s}. \tag{9}$$

Simulated data (8) were used to train GPR. Here, generalization was used because the policy parameters that match the given queries cannot be easily computed. To compute them, one would need to run an optimization procedure that would determine the appropriate robot arm velocities so that the ball is detached from the hand (remember, it is not held firmly) at an appropriate point and with an appropriate velocity. Moreover, the arm is part of the humanoid robot body, which might also move during the throwing motion so that the robot remains stable. All these effects are difficult to model and one might need RL to produce each throwing motion. That, on the other hand, would take a very long time for the 625 targets even in simulation. Nonetheless, given the relatively large amount of data in $\mathcal{D}^{\text{AE}}$, the generalized trajectories produce accurate simulated throws that are very similar to the simulated ones. Thus, using the data computed by Equation (9), we formed the dataset

$$\mathcal{D}^s = \left\{ \boldsymbol{\theta}_j^{\text{CDMP},s}, \boldsymbol{q}_j^g \right\}_{j=1}^{625}, \tag{10}$$

which we call $\mathcal{D}^s$ the *simulated* database.

The last database was generated by first programming 25 robot throwing movements with parameters $\boldsymbol{\theta}_j^r$ and executing them with a real robot. The locations $\boldsymbol{q}_j^r$ where the ball landed were measured and we formed the dataset of real robot throws

$$\mathcal{D}^r = \left\{ \boldsymbol{\theta}_j^r, \boldsymbol{q}_j^r \right\}_{j=1}^{25}. \tag{11}$$

We then applied GPR trained with data specified in (11) to compute new throws for the same locations $q_j^g$ as used when generating the simulated database (10)

$$\mathbf{G}(\mathcal{D}^r) : q_j^g \mapsto \boldsymbol{\theta}_j^{\text{CDMP},g}. \tag{12}$$

Using the parameters computed by applying Equation (12), we formed the dataset

$$\mathcal{D}^g = \left\{ \boldsymbol{\theta}_j^{\text{CDMP},g}, q_j^g \right\}_{j=1}^{625}. \tag{13}$$

We call $\mathcal{D}^g$ the *generalized* database. Note that $\mathcal{D}^s$ and $\mathcal{D}^g$ consist of the same targets $q_j$ but different throwing policy parameters. The throwing policies in $\mathcal{D}^s$ and $\mathcal{D}^g$ were generated computationally and were never executed with the real robot. The aim was to assess the effectiveness of such an approach for database collection. The targets of the autoencoder database, real-world throws, the simulated, and generalized databases are depicted overlaid in Figure 4.

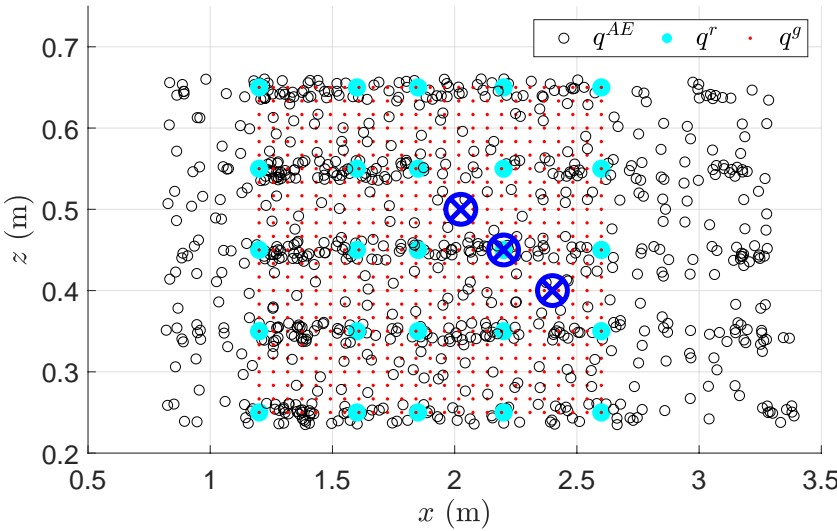

**Figure 4.** Targets contained in $\mathcal{D}^{\text{AE}}$ (black circles), real-world database $\mathcal{D}^r$ (cyan dots), and generalized databases $\mathcal{D}^r$ and $\mathcal{D}^g$ (red dots). The three blue crossed circles depict the queries which served as starting points for RL in latent space, as described in Section 4.2.

### 3.4. Evaluation Metrics

Several metrics were used for evaluation of throwing trajectories. To evaluate the error of the autoencoder network, we used mean squared error (MSE) for positions, orientations, and for the difference between the policy parameters (CDMP weights). As in our experiments the position has two degrees of freedom, MSE for position is defined as

$$\text{MSE}_{\text{pos}} = \frac{1}{K} \sum_{k=1}^{K} \frac{1}{n_k} \sum_{i=1}^{n_k} \| [x_{i,k}, z_{i,k}]^{\text{T}} - [\widetilde{x}_{i,k}, \widetilde{z}_{i,k}]^{\text{T}} \|, \tag{14}$$

where $[x_{i,k}, z_{i,k}]^{\text{T}}$ is the position at sample $i$ on the trajectory generated by the $k$-th input parameters of AE, $[\widetilde{x}_{i,k}, \widetilde{z}_{i,k}]^{\text{T}}$ the position at sample $i$ on the trajectory generated by the $k$-th output parameters of AE, $n_k$ the number of time samples on the $k$-th trajectory, and $K$ the number of test trajectories. In order to obtain the same number of samples on the trajectory generated by input and output parameters of the autoencoder, the integration time step of the output trajectory was appropriately scaled: $\widetilde{\Delta t} = (\widetilde{\tau}/\tau)\Delta t$. As in our experiments

the orientation had only one degree of freedom, we only take into consideration the angle around the $y$ axis, here denoted by $\varphi$:

$$\text{MSE}_\varphi = \frac{1}{K} \sum_{k=1}^{K} \frac{1}{n_k} \sum_{i=1}^{n_k} ||\varphi_{i,k} - \widetilde{\varphi}_{i,k}||. \tag{15}$$

For the difference in policy parameters, we used the mean squared error of policy parameter vectors without the duration $\tau$, i.e., $\boldsymbol{\theta}'_k = \boldsymbol{\theta}_k^{\text{CDMP}} \backslash \tau_k$

$$\text{MSE}_{\boldsymbol{\theta}} = \frac{1}{K} \sum_{k=1}^{K} ||\boldsymbol{\theta}'_k - \widetilde{\boldsymbol{\theta}}'_k||. \tag{16}$$

To compare the simulated and the generalized databases we used the error of the throw

$$e = ||\boldsymbol{q}_d - \boldsymbol{q}_a||, \tag{17}$$

where $\boldsymbol{q}_d$ is the desired target landing spot (location of the basket) and $\boldsymbol{q}_a$ the actual landing spot of the throw. The error metric defined in (17) was used to evaluate the throwing motions computed by GPR or by the selected RL algorithm, which in our case was reward-weighted policy learning with importance sampling [18].

## 4. Results

### 4.1. Effect of Database Size and Latent Space Dimension on the Quality of AE Approximation

We first tested the effect of the latent space dimension and the number of database entries on the training of the autoencoder network. The accuracy of the trained network was evaluated by varying the amount of training data and the dimensionality of the latent space ($L$). Database defined by Equation (8) was used for this purpose. We trained $M = 10$ (determined empirically) networks for each tested pair of training data and dimensionality of the latent space. For the training data, an appropriately sized random subset of database given by Equation (8) was chosen for each of the $M$ networks. Additionally, each of the $M$ networks was trained with a different random initialization of the parameters and tested using Equations (14)–(16) with $K = 40$ randomly selected throwing trajectories from the database specified in Equation (8), which were not included in the training dataset. Thus, we obtained $M$ results describing the error of each autoencoder network: $\text{MSE}_{\text{pos},m}, \text{MSE}_{\varphi,m}, \text{MSE}_{\boldsymbol{\theta},m}, m = 1, \dots, M$, all calculated according to Equations (14), (15), and (16), respectively. The final evaluation results were then obtained by averaging

$$\overline{\text{MSE}}_{\text{pos}} = \frac{1}{M} \sum_{m=1}^{M} \text{MSE}_{\text{pos},m}, \tag{18}$$

$$\overline{\text{MSE}}_{\varphi} = \frac{1}{M} \sum_{m=1}^{M} \text{MSE}_{\varphi,m}, \tag{19}$$

$$\overline{\text{MSE}}_{\boldsymbol{\theta}} = \frac{1}{M} \sum_{m=1}^{M} \text{MSE}_{\boldsymbol{\theta},m}. \tag{20}$$

Figure 5 shows the error of autoencoder trajectory encoding. The left plot shows the difference in policy parameters, while the right-two plots show the errors for position and orientation, as defined in Equations (20), (18), and (19), respectively. By comparing the three graphs, we can notice that the differences in quality of approximation are most pronounced in the left-most graph, that is, in the approximation of CDMP parameters. A significant decrease in error with both increasing database size, i.e., training samples, and latent space dimension can be observed. A sort of plateau is roughly reached with $L = 4$ and 300 training samples, and the error only marginal decreases afterwards. Therefore, we used these values in our next experiments. Note also that the error in the estimation of the

position trajectory, shown in the center plot of Figure 5, is slightly larger than 1 cm, whereas the error in the orientation trajectory (right plot) is about 3 degrees.

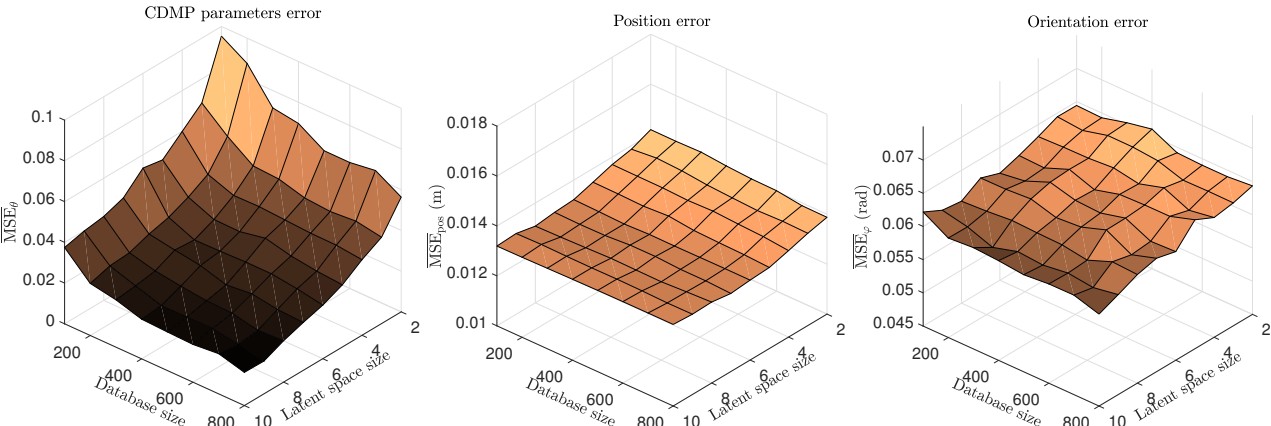

**Figure 5.** Mean squared error of AE approximation for position (**left**), orientation around the $y$ axis (**center**), and Cartesian space dynamic movement primitive (CDMP) parameters (**right**), all with respect to the amount of training data and dimensionality of the latent space.

### 4.2. RL in AE Latent Spaces

We first tested whether RL in a low-dimensional latent space defined by an autoencoder is advantageous compared to RL in the full CDMP parameter space, which has a much higher dimension. We also evaluated the performance of RL in AE latent space computed from the generalized dataset (13) compared to RL in the latent space computed from the simulated dataset (10). Note that in each RL trial, the current policy parameters are modified by adding random noise. However, the full CDMP parameter space contains all smooth movements in the Cartesian coordinate system, not just the throwing movements. Thus, the modification of policy parameters in the CDMP space can result in movements that are not throwing movements, including movements that are not safe for the robot. On the other hand, a low-dimensional latent space typically contains only throwing movements; thus, if we first project the current CDMP parameters to the latent space using Equation (2), add random noise to the projected parameters, and backproject the modified autoencoder policy parameters to the CDMP parameter space using Equation (3), we are much more likely to obtain a valid throwing movement that is safe for the robot to perform. This is how RL in latent spaces is carried out.

The comparison of RL in CDMP parameter space and AE parameter space was carried out in simulation, while the comparison between RL in AE spaces trained with either the simulated or the generalized dataset was performed in real robot experiments. Having determined the appropriate dimension of the latent space in Section 4.1, we used $L = 4$ to train all autoencoder networks. For RL testing, we applied reward-weighted policy learning with importance sampling method [18].

The first experiment was conducted in simulation in order to compare the performance of RL in the full CDMP parameter space and in AE latent space when learning to throw at 25 randomly selected targets. The initial throw to start the learning process was always the same and a "hit" was reported if the ball landed within 0.09 m of the target. We analyzed the evolution of the error of throwing for RL in the full CDMP parameter space and the AE latent space, where the error is given as the distance to the target at the height of the target. The results are shown in Figure 6. The faster convergence of learning in autoencoder latent space is evident. The left top plot in Figure 7 shows the average number and standard deviation of iterations required to compute accurate throws. Note that learning in the CDMP parameter space takes longer on the average, but the standard deviation is quite high in both cases. The one-way analysis of variance (ANOVA) between the two processes confirms that learning in latent space is statistically significantly faster ($p = 0.0457$) than

learning in the CDMP space. The right top plot in Figure 7 verifies the benefits of using AE latent space over full CDMP space by showing the worst case of 25 learning iterations. The results presented in Figures 6 and 7 are consistent with the literature where it has been reported that learning is faster in a low-dimensional latent space [5]. However, more experiments are needed to confirm that learning in AE latent spaces is faster also in general.

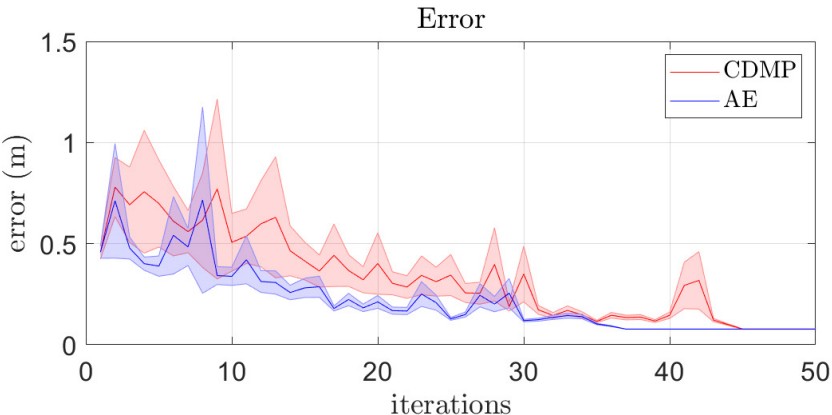

**Figure 6.** [SIM] Error of throwing and its variance for 25 instances of RL in AE latent space and CDMP space.

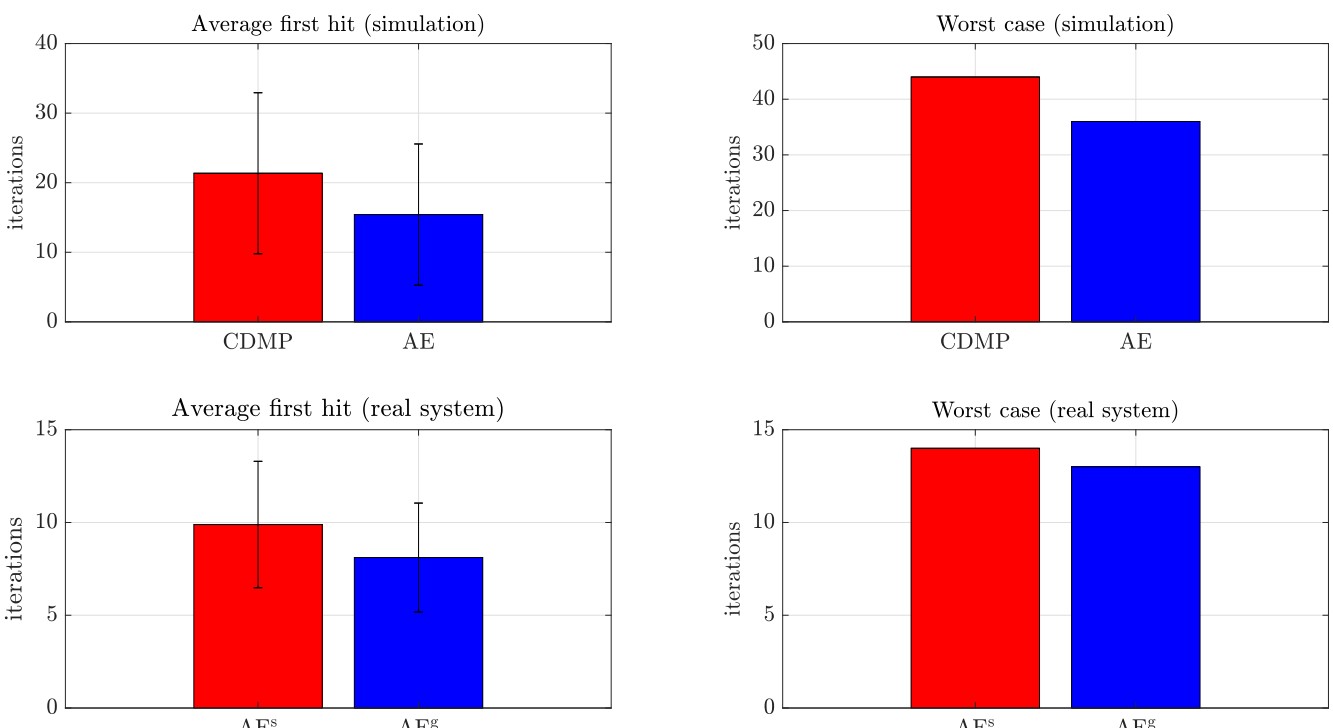

**Figure 7.** (**Left top**): [SIM] Average number and standard deviation of iterations required to hit a target for RL in AE latent space and in CDMP space. (**Right top**): [SIM] The highest number of iterations required to hit the target. All top row results were obtained in simulation. (**Left bottom**): [REAL] Average number and standard deviation of RL iterations required to hit a target in AE latent space where the AE was trained with either the simulated (red) or generalized (blue) dataset. (**Right bottom**): [REAL] The highest number of iterations required until a hit.

In the next experiment, we used a real robot to compare the performance of learning in autoencoder spaces computed using two different datasets: simulated dataset (10) and generalized dataset (13). Figure 8 shows a sequence of still images depicting a successful throw with the humanoid robot TALOS, which was used for this experiment. Three targets were used to test the learning of throwing, marked with blue, crossed circles in Figure 4.

For each of the targets, we performed RL three times in both types of latent spaces. In order to have a good initial policy, and therefore reduce the number of RL iterations, we first generated throwing trajectories for each of these targets using GPR in the CDMP space and then performed RL in both AE spaces to refine the throw and hit the given targets.

Figure 9 shows the results for real-system learning in autoencoder latent spaces trained on simulated and generalized data, respectively. A somewhat faster convergence is observed when using the generalized database, with at most 14 shots needed for learning in latent space associated with the simulated dataset, and at most 13 for the generalized dataset. The real-system results shown in the bottom plots of Figure 7 confirm faster convergence of RL in $AE^g$ even when the conditions were most favorable for the simulated database—that is, in the area where the generalization using the simulated database performed most accurately. The targets were all in the central range of the target area, where generalization already produces throws with low error. Consequently, the required number of trials was small and the difference between the applied autoencoder latent spaces was not statistically significant ($p > 0.05$). It should be noted, though, that larger exploration noise was required to achieve convergence when learning in the latent space trained on the simulated dataset. This confirms that the generalized dataset represents the actual state of the world more faithfully.

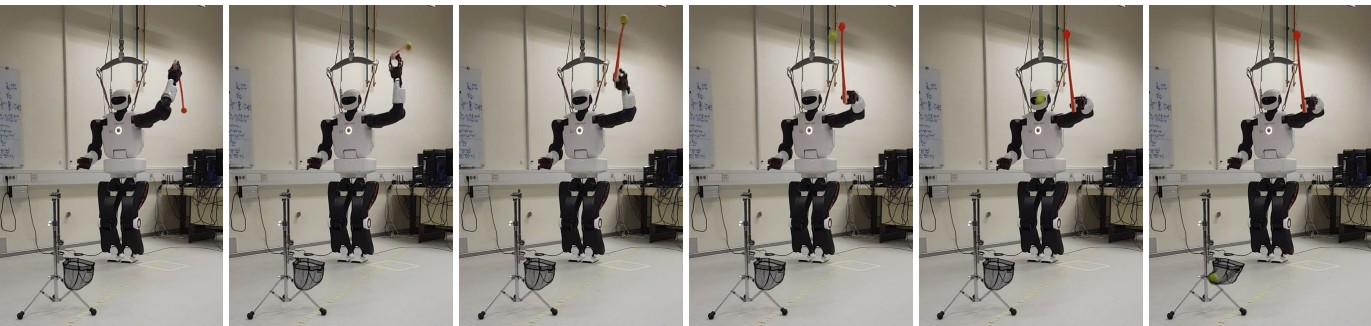

**Figure 8.** Image sequence of executing a learned throw at a target on with the TALOS robot.

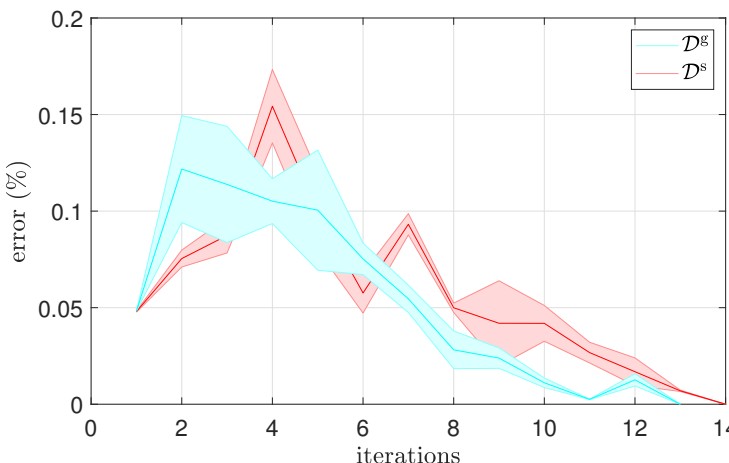

**Figure 9.** [REAL] Convergence of RL in AE latent spaces computed from either the simulated or the generalized dataset.

## 5. Conclusions

Efficient data representation with autoencoders, as has already been discussed in the literature [5,17,30,31], can reduce the dimension of the learning space. In our experiments, we have shown the effectiveness of dimensionality reduction for RL of motor skills such as throwing.

It is clear that learning of autoencoders requires the acquisition of a sufficient amount of data. This is often challenging and time-consuming if data acquisition takes place in the real world. Simulation is the most obvious substitute for real-world experiments. Still, no matter how good, simulation can never fully replicate the physics of the real world [46]. In this paper, we propose an alternative approach, i.e., to acquire data for training autoencoders by generalizing from a small number of real robot performances of the task. Our experiments show that in the case of robotic throwing, the data obtained this way can better capture the physics of the real-world. Consequently, the Reinforcement Learning of throwing becomes faster.

The task itself determines how many samples are required for training an autoencoder network for dimensionality reduction while still maintaining an accurate representation of the motor skill. We have shown that relatively few datapoints are needed to improve the Reinforcement Learning of throwing. Other tasks might require more example executions. The proposed methodology, while only shown on a relatively straightforward example, can be applied for more complex tasks, where otherwise thousands of real-world examples might be needed.

While generating the database in simulation has been shown less effective than using a generalized database, we have shown elsewhere [18] that learning in simulation can also reduce the number of required real-world task executions. These two approaches are complementary and can benefit from each other.

In the future, we plan to combine the proposed methodology with other methods that can transfer latent space actions from one robot to another and with methods that bridge the sim-to-real gap in motor learning. The proposed methodology is the first step in this direction.

**Author Contributions:** Conceptualization, methodology, investigation, and writing—original draft preparation: Z.L., R.P., and A.G.; writing—review and editing: A.U. and A.G. All authors have read and agreed to the published version of the manuscript.

**Funding:** This research was funded by Horizon 2020 RIA ReconCycle, GA no. 871352, and program group Automation, robotics, and biocybernetics (P2-0076) supported by the Slovenian Research Agency.

**Conflicts of Interest:** The authors declare no conflicts of interest.

**Appendix A. Cartesian Dynamic Movement Primitives—CDMPs**

Cartesian space Dynamic Movement Primitive (CDMP) separately provide the position and orientation trajectories. The positions are specified in the same way as in, for example, joint space DMPs [43]. The orientations are specified by unit quaternions and therefore require different parameter computation and integration.

A CDMP does not directly depend on time. The phase system provides indirect time dependency and synchronization of position and orientation degrees of freedom along the trajectory. The phase system is defined with

$$\dot{x} = -\alpha_x x, \tag{A1}$$

where $x$ is the phase that starts at 1 and converges to 0 as the trajectory goal is reached. $\alpha_x$ is a positive constant.

The positions $\boldsymbol{p}$ and orientations $\boldsymbol{\Phi}$ are specified by

$$\nu(x)\dot{z} = \alpha_z(\beta_z(\boldsymbol{g}^p - \boldsymbol{p}) - \boldsymbol{z}) + \boldsymbol{f}_p(x), \tag{A2}$$
$$\dot{\boldsymbol{p}} = \boldsymbol{z}, \tag{A3}$$
$$\dot{\boldsymbol{\eta}} = \alpha_z(\beta_z 2\log(\boldsymbol{g}^o * \overline{\boldsymbol{\Phi}}) - \boldsymbol{\eta}) + \boldsymbol{f}_o(x), \tag{A4}$$
$$\dot{\boldsymbol{\Phi}} = \frac{1}{2}\boldsymbol{\eta} * \boldsymbol{\Phi}, \tag{A5}$$

where $g^p \in \mathbb{R}^3$ is the goal position and $g^o \in \mathbb{R}^4$ the goal orientation of the movement. The orientation is in the form of unit quaternion $\mathbf{\Phi} \in \mathbb{R}^4$. The parameters $z$, $\eta \in \mathbb{R}^3$ denote the scaled linear and angular velocity ($z = \nu(x)\dot{p}$, $\eta = \nu(x)\omega$). Additional details on quaternion operations, such as product $*$, conjugation $\overline{\mathbf{\Phi}}$, and the quaternion logarithm $\log(\mathbf{\Phi})$, see in [42]. The so-called forcing terms $f_p$, $f_o : \mathbb{R} \mapsto \mathbb{R}^3$ are given with

$$f_p(x) = D_p \frac{\sum_{i=1}^{N} w_i^p \Psi_i(x)}{\sum_{i=1}^{N} \Psi_i(x)} x, \tag{A6}$$

$$f_o(x) = D_o \frac{\sum_{i=1}^{N} w_i^o \Psi_i(x)}{\sum_{i=1}^{N} \Psi_i(x)} x. \tag{A7}$$

where the weights $w_i^p$, $w_i^o \in \mathbb{R}^3$, $i = 1, \ldots, N$, encode the positions and orientations, respectively, and $N$ is the number of radial basis functions. The weights have to be learned, for example, directly from an input Cartesian trajectory $\{p_k, \mathbf{\Phi}_k, \dot{p}_k, \omega_k, \ddot{p}_k, \dot{\omega}_k, t_k\}_{k=0}^{K}$. For $D_p$, $D_o \in \mathbb{R}^{3 \times 3}$, we can use $I$, see in [42] for further possibilities. The forcing terms are composed from a linear combination of nonlinear radial basis functions (RBF) $\Psi_i$

$$\Psi_i(x) = \exp\left(-h_i(x - c_i)^2\right). \tag{A8}$$

The RBF are centered at $c_i = \exp\left(-\alpha_x \frac{i-1}{N-1}\right)$, while their width is $h_i = \frac{1}{(c_{i+1} - c_i)^2}$, $i = 1, \ldots, N$, $h_N = h_{N-1}$ [9]. The goal position and orientation are usually set to the final position and orientation on the desired trajectory, i.e., $g^p = p_{t_K}$ and $g^o = \mathbf{\Phi}_{t_K}$. For more details and auxiliary math see in [42].

## Appendix B. Gaussian Process Regression

A Gaussian process is defined as

$$g(\mathbf{q}) \sim \mathcal{GP}\big(m(\mathbf{q}), k(\mathbf{q}, \mathbf{q}')\big), \tag{A9}$$

where $m(\mathbf{q}) = \mathbb{E}(g(\mathbf{q}))$ is the mean function and $k(\mathbf{q}, \mathbf{q}') = \mathbb{E}((g(\mathbf{q}) - m(\mathbf{q}))(g(\mathbf{q}') - m(\mathbf{q}')))$ the covariance function of the process. Let us assume that we have a set of noisy observations $\{\mathbf{q}_k, \theta_k\}_{k=1}^{m}$, $\theta_k = g(\mathbf{q}_k) + \epsilon$, $\epsilon \sim \mathcal{N}(0, \sigma_n^2)$, where $\mathcal{N}$ denotes the Gaussian distribution. In our experiments, $\theta_k$ is one of the parameters describing the motion (one of the CDMP parameters), $\mathbf{q}$ is the desired target of the throw, and $g$ is the unknown nonlinear function mapping query points to the parameters of the throwing motion. Subtracting the mean from the training data, we can assume that $m(\mathbf{q}) = 0$. If we are given a set of $m_2$ new query points $\mathbf{Q}^* = \{\mathbf{q}_k^*\}_{k=1}^{m_2}$, then the joint distribution of all outputs is given as [39]

$$\begin{bmatrix} \boldsymbol{\vartheta} \\ \boldsymbol{\vartheta}^* \end{bmatrix} \sim \mathcal{N}\left(\mathbf{0}, \begin{bmatrix} \mathbf{K}(\mathbf{Q}, \mathbf{Q}) + \sigma_n^2 \mathbf{I} & \mathbf{K}(\mathbf{Q}, \mathbf{Q}^*) \\ \mathbf{K}(\mathbf{Q}^*, \mathbf{Q}) & \mathbf{K}(\mathbf{Q}^*, \mathbf{Q}^*) \end{bmatrix}\right), \tag{A10}$$

where $\mathbf{Q} = \{\mathbf{q}_k\}_{k=1}^{m}$, $\mathbf{Q}^*$, $\boldsymbol{\vartheta} = [\theta_1, \ldots, \theta_m]^{\mathrm{T}}$, $\boldsymbol{\vartheta}^*$, respectively, combine all inputs and outputs and $\mathbf{K}(\cdot, \cdot)$ are the joint covariance matrices calculated according to the model (A9). Based on joint distribution (A10), the expected value $\bar{\boldsymbol{\vartheta}}^* \in \mathbb{R}^{m_2}$ can be calculated as [39]

$$\bar{\boldsymbol{\vartheta}}^* = \mathbb{E}(\boldsymbol{\vartheta}^* | \mathbf{Q}, \boldsymbol{\vartheta}, \mathbf{Q}^*) = \mathbf{K}(\mathbf{Q}^*, \mathbf{Q})[\mathbf{K}(\mathbf{Q}, \mathbf{Q}) + \sigma_n^2 \mathbf{I}]^{-1} \boldsymbol{\vartheta}, \tag{A11}$$

with the following estimate for the covariance of the prediction,

$$\mathrm{cov}(\boldsymbol{\vartheta}^*) = \mathbf{K}(\mathbf{Q}^*, \mathbf{Q}^*) - \mathbf{K}(\mathbf{Q}^*, \mathbf{Q})[\mathbf{K}(\mathbf{Q}, \mathbf{Q}) + \sigma_n^2 \mathbf{I}]^{-1} \mathbf{K}(\mathbf{Q}, \mathbf{Q}^*).$$

One commonly used covariance function is

$$k(\mathbf{q}, \mathbf{q}') = \sigma_f^2 \sum_{i=1}^{D_q} \exp\left(-\frac{1}{2}\frac{(q_i - q_i')^2}{l_i^2}\right), \tag{A12}$$

which results in a Bayesian regression model with an infinite number of basis functions. $D_q$ denotes the dimension of the query point space. $\sigma_n^2$, $\sigma_f^2$, and $l_i$ are the hyperparameters of the Gaussian process that need to be estimated in the training phase. See in [39] for more details.

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
