# Peer review of "Generalization-Based Acquisition of Training Data for Motor Primitive Learning by Neural Networks"

_applsci, doi:10.3390/app11031013_

Round 1
Reviewer 1 Report
General comment
This paper developed an efficient database gathering approach, reducing the search space. In the paper, simulation and real-world results with a humanoid robot TALOS are provided. The topic is quite interesting to the readers, however, in my opinion the paper has some shortcomings regarding the lacks of reviewing and comparing works on current research in this field.
Detailed comments
1) The literature review in the introduction section should be improved. Please add descriptions on current research that in this field, and explain why did you select such algorithms/methods, and why did you proposed this new methodology to improve the solution.
For example, you mentioned AE is selected because it provides a nonlinear transformation, while the PCA provides a linear transformation. However, the Kernel PCA can also provides a nonlinear transformation. I agree that the AE has good performance, but I suggest to add a bit more review work to make your work more convincing.
2) In the beginning of Section 2 (Line 72), the authors claim that 'The main contributions of this paper are the analysis of the required number of training samples for AE-based dimensionality reduction and a new methodology for obtaining samples trajectories grounded in the real world data by means of statistical generalization.'
Please add several sentences or one short paragraph to summarize the novel part of the proposed methodology, and its advantage.
3) In Section 4 Results or in conclusion, please mention a bit comparing works. Or add several sentences in the Introduction part.
Author Response
Dear Vicky Jiang, MDPI Applied Sciences Assistant Editor
We thank you and the reviewers for considering our paper and for the constructive comments. We have revised the manuscript according to the reviewers’ comments. Below please find also detailed replies to individual comments.
The comments are in black, while our replies are in green. Citations of the text from the paper are in italic.
Besides the paper we attached also a version marking the difference between the original and the revised submission, for reviewers’ convenience.
General comment
This paper developed an efficient database gathering approach, reducing the search space. In the paper, simulation and real-world results with a humanoid robot TALOS are provided. The topic is quite interesting to the readers, however, in my opinion the paper has some shortcomings regarding the lacks of reviewing and comparing works on current research in this field.
Please see our replies below.
1) The literature review in the introduction section should be improved. Please add descriptions on current research that in this field, and explain why did you select such algorithms/methods, and why did you proposed this new methodology to improve the solution.
For example, you mentioned AE is selected because it provides a nonlinear transformation, while the PCA provides a linear transformation. However, the Kernel PCA can also provides a nonlinear transformation. I agree that the AE has good performance, but I suggest to add a bit more review work to make your work more convincing.
We have added an extensive “Related Work” section that discusses dimensionality reduction, including kernel PCA, generalization as a means of database acquisition, and transfer between simulation and real-world. We explicitly state what is the main difference of our paper with respect to the cited papers.
Regarding kernel PCA, we clearly say that it can be utilized for nonlinear data dimensionality reduction, but the kernel needs to be defined. On the other hand, autoencoder neural networks work in a similar fashion, but with a sort of very complex and autonomously determined kernel. We agree that the topic is relevant, but we believe autoencoders are a much better fit for our task.
2) In the beginning of Section 2 (Line 72), the authors claim that 'The main contributions of this paper are the analysis of the required number of training samples for AE-based dimensionality reduction and a new methodology for obtaining samples trajectories grounded in the real world data by means of statistical generalization.'
Please add several sentences or one short paragraph to summarize the novel part of the proposed methodology, and its advantage.
We have added a summary and the benefits of the proposed methodology in the beginning of Section 2. It now says:
In short, this paper proposes the following methodology to improve the accuracy of robot behavior and the speed of RL convergence. We first record a small number of real-world actions and generalize to a larger set. We then use this larger set to train an autoencoder. The latent space of the autoencoder is smaller than the full action representation space, which provides for a faster convergence of reinforcement learning. Because the data for autoencoder training are grounded in a small set of real-world actions, it is also closer to real-world actions than an autoencoder trained with only simulated data. As explained above, the consequence is that we reduce the amount of real-world robot executions to acquire a dataset suitable for autoencoder training and then learn faster because RL is performed in a smaller search space defined by the latent space of the autoencoder.
3) In Section 4 Results or in conclusion, please mention a bit comparing works. Or add several sentences in the Introduction part.
We have added an extensive “Related Work” section that discusses several comparing works. Please also see our reply to your first comment.

Reviewer 2 Report
Dear authors,
The following manuscript points forward, in the opinion of this reviewer, the main positive aspects of your contribution and some suggestions that may be considered in order to improve its quality for further revisions.
The paper proposes a new method for generating efficient database by applying statistical generalization and use it during the Reinforcement Learning (RL) of a real humanoid robot TALOS in a throwing at a target task. Among the list of positive points of the work, it is remarkable the novelty and originality of the work, since it applies generalization techniques to overcome the dimensionality problem of RL algorithms in the context of real robotics. In this sense, the paper presents meaningful results both in simulated and real domains.
Considering the suggestions after the revision of the manuscript, the reviewer would like to highlight the next ideas in order to guide the authors in the improvement of its work. The following list contains both subtle and important considerations to enhance the quality of the work.
- First, the most important aspect that should be tackle is the inclusion of a section that clearly states the background and related work to the present manuscript. It its true that the introduction and included appendixes contain references to other work and techniques in the same context. Nevertheless, it may be necessary to provide a grounded basis for the work.
- Second,inside the results section, it is important to separate in the text simulated and real results. Despite the need of running the method in simulation, real trials represent a very important and powerful tool if these methods want to be replicated on real robot in the future.
- In the introduction, the term However is repeated many often. Replace it to avoid repetition.
- The beginning of Section 2 does not emphasize the topic of the section and contains a missing reference in the initial sentence.
- The paragraph starting in line 83 Section 2 “As shown in [19] ...” should be rewritten. It is difficult to understand the operation of Autoencoders NN without a previous definition and context.
- Some terms included in equations are not explained in the text, some of them (like N) are multiple defined in different equations with different context, and the value set for some of them is not argued in the text. For example, it is not reflected why the training of the AE network finishes after 60 validation steps without improvement, or why the number of weights N is set to 25 or the number of trained networks M = 10, among others.
- Some references are missing when making important assumptions.
- E.g. In line 126-127, it is said that “GPR has been shown to be effective when estimating the inverse dynamics of a seven DOF robot arm.”, but it is not referenced the origin of this statement.
- There exists some paragraphs with just 1-2 lines that must be avoided in most of the cases.
- Multiple references to Figures, Equations, and Citations are not correctly defined. It may be a possible solution to use a standard format since sometimes it is not clear if the authors are referring to a Figure or Equation. For example, Equations from line 220 are just referenced in parenthesis and may be referenced Equation 20, for example.
- Finally, the reviewer encourages the authors to reread the text in order to avoid multiple typos and spelling errors found in the text.
Hoping these lines help the authors to improve its contribution.
Best regards.

Author Response
Dear Vicky Jiang, MDPI Applied Sciences Assistant Editor
We thank you and the reviewers for considering our paper and for the constructive comments. We have revised the manuscript according to the reviewers’ comments. Below please find also detailed replies to individual comments.
The comments are in black, while our replies are in green. Citations of the text from the paper are in italic.
Besides the paper we attached also a version marking the difference between the original and the revised submission, for reviewers’ convenience.
Dear authors,
The following manuscript points forward, in the opinion of this reviewer, the main positive aspects of your contribution and some suggestions that may be considered in order to improve its quality for further revisions.
Please see our replies below.
The paper proposes a new method for generating efficient database by applying statistical generalization and use it during the Reinforcement Learning (RL) of a real humanoid robot TALOS in a throwing at a target task. Among the list of positive points of the work, it is remarkable the novelty and originality of the work, since it applies generalization techniques to overcome the dimensionality problem of RL algorithms in the context of real robotics. In this sense, the paper presents meaningful results both in simulated and real domains.
Considering the suggestions after the revision of the manuscript, the reviewer would like to highlight the next ideas in order to guide the authors in the improvement of its work. The following list contains both subtle and important considerations to enhance the quality of the work.
- First, the most important aspect that should be tackle is the inclusion of a section that clearly states the background and related work to the present manuscript. It its true that the introduction and included appendixes contain references to other work and techniques in the same context. Nevertheless, it may be necessary to provide a grounded basis for the work.
We have added an extensive “Related Work” section that discusses dimensionality reduction, including kernel PCA, generalization as a means of database acquisition, and transfer between simulation and real-world. We clearly state what is the main difference of our paper with respect to the cited papers.
- Second,inside the results section, it is important to separate in the text simulated and real results. Despite the need of running the method in simulation, real trials represent a very important and powerful tool if these methods want to be replicated on real robot in the future.
We have thoroughly rewritten a part of the Results Section, specifically Section 4.2: RL in AE latent spaces , as requested. Now, we first describe the simulated results, followed by real-world results. Figure captions for relevant Figs. 6, 7 and 8 also clearly indicate which results are from simulation and which from real-world experiments.
- In the introduction, the term However is repeated many often. Replace it to avoid repetition.
We have replaced several instances of the term to make it less frequent. However, we left some of them in the paper.
- The beginning of Section 2 does not emphasize the topic of the section and contains a missing reference in the initial sentence.
We have expressed more clearly that this is the methodology section, and that its content is used to derive the main contributions of the paper. We have also added a short summary and the benefits of the proposed methodology.
- The paragraph starting in line 83 Section 2 “As shown in [19] ...” should be rewritten. It is difficult to understand the operation of Autoencoders NN without a previous definition and context.
We have rewritten this and the next paragraph to make clear what is an autoencoder, what is the latent space, what are the latent space parameters, how they are determined from an input data using the tanh neuron activation function, and how the network is trained. We believe that these paragraphs are now much clearer.
- Some terms included in equations are not explained in the text, some of them (like N) are multiple defined in different equations with different context, and the value set for some of them is not argued in the text. For example, it is not reflected why the training of the AE network finishes after 60 validation steps without improvement, or why the number of weights N is set to 25 or the number of trained networks M = 10, among others.
We have now changed N to N_s in equations (1), (4) and (5). Please note that the variable N_s represents the number of training samples, and each training sample is given by a set of parameters that define a robot trajectory. Thus, N_s is the same in (1), (4) and (5). Only the number of DMP kernel functions in (A6) and (A7) is now marked with N. Thank you for noticing this obvious mistake.
The training of neural networks was stopped after 60 validations without improvement, because obviously at some point the training gets as good as it ever will. This number 60 was determined based on previous experience, i.e., empirically. It says now in the paper at the end of paragraph after Eq. (1):
The results are then averaged. The training of each autoencoder network is terminated if the results do not improve in 60 consecutive validation steps. This number was determined empirically.
The number of DMP kernel functions was set to 25 as this was the number typically used because it provided sufficient accuracy. It says in the paper:
If, for example, the number of weights is N = 25 (this is usually sufficient for good accuracy, see [9]), we obtain a 165 dimensional parameter space.
The number of trained networks was set to 10 empirically. We now say it in the paper.
- Some references are missing when making important assumptions.
- g. In line 126-127, it is said that “GPR has been shown to be effective when estimating the inverse dynamics of a seven DOF robot arm.”, but it is not referenced the origin of this statement.
We have now cited this statement with the reference:
Williams, C.; Klanke, S.; Vijayakumar, S.; Chai, K. Multi-task Gaussian Process Learning of Robot Inverse Dynamics. Advances in Neural Information Processing Systems, 2009, pp. 265–272.
We have also added a comprehensive “Related Work” section after the introduction, please also see our reply to your first comment..
- There exists some paragraphs with just 1-2 lines that must be avoided in most of the cases.
We removed some of the short paragraphs, as requested, but left some at places where we think they emphasize their messages.
- Multiple references to Figures, Equations, and Citations are not correctly defined. It may be a possible solution to use a standard format since sometimes it is not clear if the authors are referring to a Figure or Equation. For example, Equations from line 220 are just referenced in parenthesis and may be referenced Equation 20, for example.
It is standard practice, if not convention, that in scientific papers mathematical equations are referenced with parentheses (the only exception are the equations at the beginning of sentences). When saying “(8)”, this clearly indicates “Eq. (8)”. Please consider this analogical example. When we say: “See [38] for more details”, it is clear that [38] refers to a reference, without the need to explicitly say “reference [38]”. The same is valid for equations and parentheses. We believe that our text, where it refers to various databases, is the most straightforward and clear if we say: “Database (8) … “, which clearly indicates that this is a database defined with equation (8).
We also checked that we strictly use “Fig. 3”, or “Figure 3” if it is at the start of a sentence to reference to figures. The numbers “3” and “8” are, of course, just examples.
- Finally, the reviewer encourages the authors to reread the text in order to avoid multiple typos and spelling errors found in the text.
We have thoroughly checked the paper and hopefully we removed the typos and spelling errors.
Hoping these lines help the authors to improve its contribution.
Thank you.

Round 2
Reviewer 1 Report
Thanks to the introduced changes, the authors' motivations for the work done became more obvious, which was missing in the first version. I recommend this article to be accepted.
Reviewer 2 Report
Dear authors,
Thank you very much for your consideration in reviewing the paper according to my suggestions. In my opinion, the inclusion of a new section setting the background of the work and the reorganization of the results endows the paper with more perspectives and a feeling of homogeneity.
Next, you can find my review to the last version of the manuscript. Note that these comments just suppose subtle suggestions for the final version.
- In the introduction, in line 30, the object “real world” can be omitted in the sentence, it sounds repetitive.
- In line 34, the task of “reaching” sounds weird, it may be describe in more detail the task of “reaching something”. Besides, in the next sentence it its defined reaching from the left or right. Maybe the proper term is approaching instead of reaching.
- The term Reinforcement Learning, when describing the technique, must be in capitals. Additionally, as the abbreviation RL is defined, use it when possible.
- In line 47 of the introduction the expression “curse of dimensionality” should be “course of dimensionality”.
- In line 48, the new paragraph has a missing point symbol at the end of the sentence.
- In line 78, the correct sentence should be “… with a TALOS robot ...”.
- In Section 2, please rewrite the sentence “Both these works list obtaining a suitable database and working with high amounts of data among the key problems.”, it is difficult to understand.
- In Section 2, there is a missing statement in lines 83-85 after the connector and.
- Please rewrite the second paragraph of Section 2.1, specially when the word “yet” is included.
- Replace the line 178 containing the sentence “Obtaining such a an amount of samples with a robot is time consuming job that causes too much wear and tear of the equipment and might even damage the robot itself.” by Obtaining such a an amount of samples with a robot is a time consuming job that causes too much wear and tear of the equipment and might even damage the robot itself.”.
- The reference describing the TALOS robot (40), should be moved before (Introduction), next to the first time the robot is presented.
- In line 310, the word “Cartesian” should not be in capitals.
- Rewrite the expression in line 399, it is difficult to understand.
Best regards and good luck in future endeavors,
Marcos.
